# NGS Evaluation of a Bernese Cohort of Unexplained Erythrocytosis Patients

**DOI:** 10.3390/genes12121951

**Published:** 2021-12-04

**Authors:** Katarzyna Aleksandra Jalowiec, Kristina Vrotniakaite-Bajerciene, Annina Capraru, Tatiana Wojtovicova, Raphael Joncourt, Alicia Rovó, Naomi A. Porret

**Affiliations:** Department of Hematology and Central Hematology Laboratory, Inselspital, University Hospital Bern, University of Bern, 3008 Bern, Switzerland; kristina.vrotniakaite-bajerciene@insel.ch (K.V.-B.); annina.capraru@insel.ch (A.C.); tatiana.wojtovicova@insel.ch (T.W.); Raphael.Joncourt@insel.ch (R.J.); Alicia.Rovo@insel.ch (A.R.); NaomiAzur.Porret@insel.ch (N.A.P.)

**Keywords:** erythrocytosis, polycythemia, NGS

## Abstract

(1) Background: Clinical and molecular data on patients with unexplained erythrocyto-sis is sparse. We aimed to analyze the clinical and molecular features of patients with congenital erythrocytosis in our tertiary reference center. (2) Methods: In 34 patients with unexplained erythrocytosis, a 13-gene Next-Generation Sequencing erythrocytosis panel developed at our center was conducted. (3) Results: In 6/34 (18%) patients, eight different heterozygous gene variants were found. These patients were, therefore, diagnosed with congenital erythrocytosis. Two patients had two different gene variants each. All variants were characterized as variants of unknown significance as they had not previously been described in the literature. The rest of the patients (28/34, 82%) had no detected gene variants. (4) Conclusions: Our experience shows that the NGS panel can be helpful in determining the reasons for persistent, unexplained erythrocytosis. In our cohort of patients with erythrocytosis, we identified some, thus far unknown, gene variants which may explain the clinical picture. However, further investigations are needed to determine the relationship between the molecular findings and the phenotype.

## 1. Introduction

Erythrocytosis is defined by an increase in hemoglobin (Hb) concentration, hematocrit (Hct) or red blood count (RBC) above the reference range adjusted to age, sex and living altitude.

Causes of erythrocytosis can be both congenital and acquired. Congenital erythrocytosis can be either associated with reduced P50 (partial pressure of oxygen at which 50% of hemoglobin is saturated with oxygen), as in high-oxygen-affinity hemoglobinopathy, or with normal P50, as in Von Hippel–Lindau syndrome (*VHL*), Egl-9 family hypoxia inducible factor 1 (*EGLN1*), also called prolyl hydroxylase domain 2 (*PHD2*), endothelial PAS domain-containing protein 1 (*EPAS1*), also known as hypoxia-inducible factor-2alpha (*HIF-2alpha*), or erythropoietin receptor (*EPOR*) mutated patients. Other genes, such as those encoding Bisphosphoglycerate Mutase (*BPGM*), Hemoglobin Subunit β (*HBB*), Hemoglobin Subunit α 1 (*HBA1*), Hemoglobin Subunit α 2 (*HBA2*), Hypoxia Inducible Factor 3 Subunit α (*HIF3A*) and SH2B Adaptor Protein 3 (*SH2B3*) have also been described in the literature [1,2].

Acquired erythrocytosis can be attributed to a clonal process called polycythemia vera (PV), in which the erythrocytosis is independent of the mechanisms that normally regulate erythropoiesis. Alternatively, it can be attributed to a secondary cause, resulting in tissue hypoxia due to a variety of conditions, including chronic lung disease, right-to-left cardiopulmonary shunts, living at high altitudes, tobacco use, sleep apnea and hypoventilation or renal artery stenosis, with the erythrocytosis being hypoxia-driven in those cases. In contrast to hypoxia, there are secondary erythrocytosis-independent processes such as those observed during the use of androgen preparations, the history of renal transplantation, and tumors such as cerebellar hemangioblastoma, meningioma, pheochromocytoma, uterine leiomyoma parathyroid adenoma, hepatocellular carcinoma, renal cell carcinoma, and renal cysts [3].

PV is a type of myeloproliferative neoplasia (MPN) characterized by erythropoietin (EPO) independent erythropoiesis. It has been shown that growing erythroid colonies in vitro from patients with PV does not require the addition of exogenous EPO [4]. The hallmark of PV is a somatic Janus kinase 2 (*JAK2*) *V617F* mutation [5,6,7], leading to constitutive hyperactivation of the JAK/STAT pathway. Further, a *JAK2* mutation in *exon 12* was observed in patients lacking the *JAK2 V617F* mutation [8]. *JAK2 V617F* and *JAK2 exon 12* mutations are responsible for over 95% and 2–3% of all PV cases [9], respectively. 

The 2016 revision of the World Health Organization classification of myeloid neoplasms resulted in an updated diagnostic criteria for PV [10]. 

Diagnosis of PV has major therapeutic consequences as these patients are at high risk of thromboembolic events and, therefore, their hematocrit must be strictly kept under 45% to avoid complications. Treatment depends on age and history of thrombosis, which defines the risk and proper disease management. The treatment includes venesections, platelet aggregation inhibitors and cytoreductive therapy, the latter being used only in high-risk cases [11].

The evaluation of patients with erythrocytosis is primarily focused on the determination of the process as either clonal or secondary. A clinical history should be acquired in order to determine a potential secondary cause. If a clonal process is suspected, laboratory work-up is required and implies testing for the *JAK2 V617F* mutation and EPO. In contrast to PV, where EPO is mostly suppressed, in secondary erythrocytosis EPO is mostly normal or increased. In the event of a negative *JAK2 V617F* mutation and suppressed EPO, further testing to confirm the *JAK2* exon 12 mutation is recommended. If none of these mutations are confirmed, *EPOR* mutations should be investigated. If EPO is normal or increased, testing of P50 is helpful to differentiate hypoxia-driven erythrocytosis [12]. 

In patients with polycythemia without a confirmed *JAK2* mutation or underlying secondary cause, further investigations are challenging and not routinely conducted. Frequently, these patients are classified as having unexplained erythrocytosis, which may be bothersome for them as well as for clinicians, since this nomination does not have a clear clinical meaning. Frequently, these patients are treated as a PV and subjected to venesections or sometimes even cytoreductive therapy. Currently, the benefits or disadvantages of such treatments in *JAK2* negative patients remain unclear and are not routinely recommended by guidelines [13]. 

Next-generation sequencing (NGS) can be used to determine a number of causes of unexplained erythrocytosis [1,2] and may be conducted, in selected cases, before a bone marrow examination. The use of NGS in investigating erythrocytosis allows for the rapid and cost-effective examination of multiple gene mutations known as underlying causes of the clinical picture. 

In the present study, we aimed to analyze the clinical and molecular features of patients with congenital erythrocytosis in our clinic. At our tertiary reference center, we developed an NGS panel to investigate the causes of *JAK2* negative erythrocytosis. In this paper, the NGS panel is described, as well as the findings in patients in whom it was conducted, especially focusing on those in whom a mutation was detected.

## 2. Materials and Methods

Between 2019 and 2021, 34 patients with unexplained erythrocytosis, defined according to WHO 2019 diagnostic criteria for PV (Hemoglobin >16.5 g/dL in men and >16 g/dL in women, or hematocrit >49% in men and >48% in women, or red cell mass >25% above mean normal predicted value), were clinically referred to investigate the underlying cause. All patients were *JAK2 V617F* as well as *JAK2* exon 12 mutation negative and had no identified secondary causes of erythrocytosis. A 13-gene NGS panel was investigated in all of them. The NGS panel analyzed genes included *EPOR* (exons 7, 8), *VHL* (orf including exon 1′), *EGLN1, EPAS1, EPO* (including several regulatory regions), *JAK2* (exons 9–16), *BPGM, HBB, HBA1, HBA2, HIF3A, OS9,* and *SH2B3* (somatic). Only gene variants considered to be pathogenic, likely pathogenic or variants of unknown significance (VUS), according to ACMG guidelines 2015, [14] are mentioned. All mentioned variants were confirmed by Sanger sequencing.

## 3. Results

The median age of the patients in whom the NGS panel was conducted was 34 (range 17–80) and the median Hb and Hct were 178 g/L (range 162–207) and 51% (range 42–60), respectively. The majority of patients were males (32/34, 94%) which is in line with previous studies showing that males suffer more from *JAK2* negative polycythemia [15]. In 6/34 (18%) patients, eight different heterozygous gene variants were found (Table 1). These patients were, therefore, diagnosed with congenital erythrocytosis. Two patients had two different mutations each. All mutations were characterized as VUS, and they have not been previously described in the literature. However, these findings in patients with erythrocytosis suggest a pathogenic likelihood. The rest of the patients (28/34, 82%) had no detected mutations.

Patient 1 was 32 years old at erythrocytosis diagnosis and his Hb and Hct values were 181 g/L and 54% respectively, consistent in several subsequent determinations. He had no history of thromboembolic events and no positive family history. Despite extensive investigations including EPO level, which was normal, stem cell cultures, bone marrow examination, and erythrocyte and plasma volume measurement, which showed absolute erythrocytosis and polysomnography, no cause for erythrocytosis was identified. Thus, he was treated with venesections. Eleven years after the initial diagnosis, the NGS panel was conducted and showed a heterozygous variant in the *EPO* gene, c.*656G > A. The variant is not listed in the following databases: HGMDprofessional, ClinVar, LOVD, and gnomAD. The *EPO* c.*656G > A variant is present in the 3′ regulatory region and is outside of the coding sequence. This variant is very rare, and it is possible that it could have an impact on the regulation of the *EPO* gene [16]. The applied ACMG criteria is PM2, for its rarity, and BP7, since it is not located in a coding region or region with splice-altering consequence, resulting in a classification of VUS.

Patient 2 was 34 years old at erythrocytosis diagnosis and his Hb and Hct values were 195 g/L and 57%, respectively. He also had no history of thromboembolic events and no positive family history. Despite extensive investigations including EPO level, which was normal, stem cell cultures, bone marrow examination, erythrocyte and plasma volume measurement, which showed absolute erythrocytosis, whole body computer tomography and head magnetic resonance imaging, no cause for erythrocytosis was identified. He was treated with venesections and developed iron deficiency anemia. Six years on from the initial diagnosis, the NGS panel was conducted and showed a heterozygous variant in *EGLN1* c.1088T > G, p.(Leu363Arg). Mutations in *EGLN1* gene cause erythrocytosis with increased or normal EPO levels, and their inheritance is autosomal dominant [17]. The variant detected here is not listed in the following databases: HGMDprofessional, ClinVar, dbSNP, LOVD, gnomAD (ACMG criterion PM2). It can, however, be located in the catalytic domain of the EGLN1 protein spanning amino acids 291–392 (ACMG criterion PM1). Computational prediction tools indicated a deleterious effect (ACMG criterion PP3). The variant was evaluated as likely pathogenic according to the ACMG criteria of PM1, PM2, PP2 and PP3. There are known variants causing erythrocytosis in this region, although they are rare. 

Patient 3 was 80 years old at erythrocytosis diagnosis and his Hb and Hct values were 207 g/L and 60% respectively. His EPO level was normal and a measurement of erythrocyte to plasma volume was not conducted. He presented with massive bilateral pulmonary embolism and suffered from a non-ST-segment elevation myocardial infarction 3 months after. He was treated with venesections after the cardiac event. Six months on from the initial presentation, the NGS panel was conducted and showed a heterozygous variant in *EGLN1,* c.122_124delACT, p.(Tyr41del). Pathogenic variants in the *EGLN1* gene cause familial erythrocytosis type 3 (ECYT3, MIM 609820) with autosomal dominant inheritance. The variant detected here leads to a deletion of an amino acid (ACMG criterion PM4) which resides in the functionally important zinc finger domain, spanning amino acids 21–58 (ACMG criterion PM1). Previously, a Y41C variant associated with erythrocytosis was reported to impair zinc finger function [18]. Consequently, the variant detected here was classified as likely pathogenic (with criteria PM1, PM2, PM4). 

Patient 4 was 20 years old at erythrocytosis diagnosis and his Hb and Hct values were 192 g/L and 56% respectively. He had no history of thromboembolic events and no positive family history. Despite extensive investigations including EPO level, which was normal; stem cell cultures; bone marrow examination; erythrocyte and plasma volume measurement, which showed absolute erythrocytosis, no cause for erythrocytosis was identified. As a result, he was treated with a venesection once. Seven years from the initial diagnosis, the NGS panel was conducted and showed a heterozygous variant in the *VHL* gene, c.340 + 648T>C. The variant is relatively common (gnomAD minor allele frequency 0.0131) but the homozygote count of two was less than the threshold of three for this recessive gene, and, therefore, ACMG criterion PM2 was applied. Therefore, it was classified as VUS. The detected variant *VHL* c.340 + 648T>C is in exon 1′, which is caused by alternative splicing, and has been described for VHL patients [19]. Because the gene variant in this case was heterozygous and no second variant was detected, it is uncertain whether this was causative of the clinical picture, whether a second variant was present in uncovered areas of the gene (e.g., in the promoter region), or if it was undetectable with the used method (e.g., large insertions or deletions). 

Patient 5 was 22 years old at erythrocytosis diagnosis and his Hb and Hct values were 170 g/L and 48%, respectively. He had no history of thromboembolic events and no positive family history. A membrane RBC disorder of stomatocytosis was detected with ektacytometry. His EPO level was normal, so a measurement of erythrocyte to plasma volume was not conducted. Ten years after the initial erythrocytosis diagnosis, the NGS panel was conducted and showed two different heterozygous variants in the *JAK2* gene c.1711G>A p.(Gly571Ser), and in the *EPAS1* gene, *c.466G>T, p.(Gly156Trp).* The *JAK2* variant is rare with a gnomAD minor allele frequency of 0.000471 (ACMG criterion PM2). Computational tools predicted a pathogenic effect (PP3). Pathogenic *JAK2* variants can cause hereditary thrombocytosis and hereditary erythrocytosis with autosomal dominant inheritance. The *JAK2* c.1711G>A, p.(Gly571Ser) variant detected here has already been described in the literature, not as clearly pathogenic, but in a somatic context in MPN [20,21,22,23]. In vitro assays did not reveal any changes in activity for this variant compared to the wild type [23] (ACMG criterion BS3). The clinical significance of this variant is therefore unclear. It was seen in triple negative MPN, and lies in the JH2 inhibitory domain, possibly leading to constitutive JAK2 signaling, but with reduced penetrance as shown in the literature. In the *EPAS1* (NM_001430.5) gene, the heterozygous variant c.466G>T, p.(Gly156Trp) was detected. This variant is rare (gnomAD minor allele frequency: 0.0000319) and computational tools predict a pathogenic effect (PP3). Pathogenic *EPAS1* variants can cause hereditary erythrocytosis with autosomal dominant inheritance. It occurs mainly in the oxygen-dependent degradation (ODD) domain in exon 12 [24]. The variant detected here lies in exon 5, just at the border of the PAS 1 domain (80–154 or sometimes 157). This is, therefore, classified as being of unclear clinical significance according to the ACMG guidelines. Generally, erythrocytosis-associated mutations in *JAK2* are often associated with low EPO, whereas those in *EPAS1* are often associated with increased EPO. However, in this patient with a normal EPO level; this laboratory test was not very helpful in distinguishing the relative importance of the two mutations observed. 

Patient 6 was 27 years old at erythrocytosis diagnosis and his Hb and Hct values were 174 g/L and 47% respectively. His EPO level was normal and erythrocyte and plasma volume measurement showed absolute erythrocytosis. He had no history of thromboembolic events and no positive family history. Seven years on from the initial erythrocytosis diagnosis, the NGS panel was conducted and showed two different heterozygous variants in *JAK2,* c.1169C>T, p.(Pro390Leu), and SH2B3, c.107C>A, p.(Ala36Glu). The *JAK2* variant, c.1169C>T, p.(Pro390Leu) detected here is unknown in the literature to date. It is very rare with only one entry in the gnomAD database (gnomAD minor allele frequency: 0.00000404, ACMG criterion PM2). Computational tools predict a deleterious effect (ACMG criterion PP3). Pathogenic *JAK2* variants can cause a type of hereditary thrombocythemia with autosomal dominant inheritance [25]. They mainly affect the two JAK homology domains in the C-terminal half of the protein. The variant c.1169C>T, p.(Pro390Leu) is located between the FERM and SH2 domains in the N-terminal half of the protein. The variant was evaluated as a VUS according to the criteria PM2 and PP3 of the ACMG guidelines. The *SH2B3* c.107C>A, p.(Ala36Glu) variant is uncovered in the existing literature. Its gnomAD minor allele frequency is 0.00146 (ACMG criterion PM2). Pathogenic *SH2B3* variants can cause hereditary erythrocytosis with autosomal dominant inheritance. They mainly affect the central PH domain [26]. The variant c.107C>A, p. (Ala36Glu) lies outside of this domain at the N-terminus of the protein. We rated this variant as a VUS according to the ACMG criterion PM2. Both genes have also been known to be responsible for somatic mutations in MPN. 

## 4. Conclusions

Our experience shows that the NGS panel can be helpful in determining the reason for persistent, unexplained erythrocytosis. In our cohort of patients with unexplained erythrocytosis, we identified some previously unknown gene variants which may explain the clinical picture. The systematic investigation of close relatives, affected or not by erythrocytosis, represents an important step towards better understanding the meaning of new variants. Indeed, analyzing trios (parents and index patient) might help in identifying other potentially relevant genes for this disease. The clinical reality shows, however, that genetic testing, in many countries, is strictly regulated by law and, therefore, the feasibility of testing relatives is doubtful even in a specialized clinic, as they need to be formally referred for the investigations. The missing data here represents a study limitation. Furthermore, despite the extensive panel performed, the majority of the patients with *JAK2 V617F* and *exon 12* negative erythrocytosis remain undiagnosed, suggesting that there might be other, still unknown, mutations causing the phenotype. Further studies addressing this relevant scientific question are needed. Apart from further development of the molecular panel, where, for example, new genes or sequencing of promoters of known genes could be added, some secondary forms may still be undiagnosed in this cohort. The results of our study are in line with previous publications from others, [1,2] where by a large portion of patients remain without a genetic explanation of their condition. 

## Figures and Tables

**Table 1 genes-12-01951-t001:** Congenital erythrocytosis on NGS panel (*N* = 6).

Patient	Age at Diagnosis	Sex	Hb (g/L), Hct (%), RBC (T/L)	EV/PV Test	EPO (mU/mL)	Gene	Accession-Nr.	Variant	dbSNP ID	Zygosity	Inheritance	ACMG Classification
1	32	M	181, 54, 5.88	Absolute erythrocytosis	16.4 (normal)	*EPO*	NM_000799.4	c.*656G>A	rs1186897562	het	AD	VUS (PM2, BP7)
2	34	M	195, 57, 5.89	Absolute erythrocytosis	8.45 (normal)	*EGLN1*	NM_022051.2	c.1088T>G p.(Leu363Arg)	N/A	het	AD	Likely pathogenic (PM1, PM2, PP2, PP3)
3	80	M	207, 60, NA	NA	7.6 (normal)	*EGLN1*	NM_022051.3	c.122_124delACT, p.(Tyr41del)	rs1182227189	het	AD	Likely pathogenic (PM1, PM2, PM4)
4	20	M	192, 56, 6.48	Absolute erythrocytosis	6.6 (normal)	*VHL*	NM_000551.4	c.340+648T>C	rs73024533	het	AR/AD	VUS (PM2)
5	22	M	170, 48, 5.48	NA	11.09 (normal)	*EPAS1*	NM_001430.5	c.466G>T, p.(Gly156Trp)	rs377257704	het	AD	VUS (PM2, PP3)
						*JAK2*	NM_004972.4	c.1711G>A, p.(Gly571Ser)	rs139504737	het	AD/somatic	VUS (PM2, PP3, BS3)
6	27	M	174, 47, 5.54	Absolute erythrocytosis	6.06 (normal)	*JAK2*	NM_004972.4	c.1169C>T, p.(Pro390Leu)	rs768074072	het	AD/somatic	VUS (PM2, PP3)
						*SH2B3*	NM_005475.3	c.107C>A, p.(Ala36Glu)	rs574829930	het	AD/somatic	VUS (PM2)

Genes tested (*N* = 13): *EPOR* (exons 7, 8), *VHL* (orf including exon 1′), *EGLN1*, *EPAS1*, *EPO* (including several regulatory regions), *JAK2* (exons 12–16), *BPGM*, *HBB*, *HBA1*, *HBA2*, *HIF3A*, *OS9*, *SH2B3* somatic. Method: Ion AmpliSeq custom gene panel, sequencing on Thermo Fisher Ion Torrent S5. Interpretation: Only gene variants of the categories pathogenic, likely pathogenic and variant of unknown significance (VUS) according to ACMG guidelines 2015 are mentioned. Abbreviations: AR: autosomal recessive, AD: autosomal dominant; EPO: erythropoietin. EV/PV: erythrocyte volume to plasma volume, F: female, het: heterozygous, M: male, VUS: variant of unknown significance.

## Data Availability

For this study, there are no publicly archived datasets analyzed or generated during the study.

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
