# Peer review of "NGS Evaluation of a Bernese Cohort of Unexplained Erythrocytosis Patients"

_genes, 2021, doi:10.3390/genes12121951_

Round 1

Reviewer 1 Report

This manuscript employs a next generation sequencing panel to investigate idiopathic erythrocytosis in 34 patients.  In six patients, 8 different heterozygous mutations were identified in the EPO, EGLN1, VHL, EPAS1, JAK2, and SH2B3 genes.  All of these genes have previously been described as causes of erythrocytosis.  The particular variants have not been described previously, and all were classified as variants of unknown significance (VUS).

This report adds to the growing literature describing mutations in erythrocytosis-associated genes.  I have the following comments.

Introduction, line 38.  I am not aware of any functional evidence that convincingly links mutations in OS9 to erythrocytosis.  I suggest deleting this gene from this sentence.

Methods, line 94.  Should “JAK2 negative” be changed to “JAK2 V617F negative”?  The authors subsequently describe two patients (patients 5 and 6) with JAK2 mutations.

Patient 2.  The L363R mutation in EGLN1 affects a residue that is buried in the catalytic domain of the protein (PMC1502536).  This nonconservative substitution is likely to affect the folding/stability of the protein.  In my opinion, this variant is better classified as likely pathogenic (with criteria PM1, PM2, PP2, PP3) as opposed to VUS.

Patient 3.  The Y41del mutation in EGLN1 deletes Tyr-41.  This amino acid resides in the zinc finger of PHD2, and a Y41C mutation associated with erthrocytosis was previously reported to impair zinc finger function (PMC6161772).  This variant should therefore be classified as likely pathogenic (with criteria PM1, PM2, PM4, PM5) as opposed to VUS.

All patients.  Are EPO values known for any of the patients?  If so, they should be reported.  This would be particularly helpful in understanding etiology in those patients with two heterozygous mutations.  For example, patient 5 has mutations in the JAK2 and EPAS1 genes.  Erythrocytosis associated mutations in JAK2 are often associated with low EPO, whereas those in EPAS1 are often associated with increased EPO.  Hence, an EPO level in this patient may help in distinguishing the relative importance of the two mutations observed.

Author Response

  • Introduction, line 38.  I am not aware of any functional evidence that convincingly links mutations in OS9 to erythrocytosis. I suggest deleting this gene from this sentence.
    • Our answer: the issue has been addressed; the gene has been deleted from this sentence.
  • Methods, line 94.  Should “JAK2 negative” be changed to “JAK2 V617F negative”?  The authors subsequently describe two patients (patients 5 and 6) with JAK2 mutations.
    • Our answer: the issue has been addressed; the correct names of the mutations in this context are JAK2 V617F and JAK2 exon 12.
  • Patient 2. The L363R mutation in EGLN1 affects a residue that is buried in the catalytic domain of the protein (PMC1502536).  This nonconservative substitution is likely to affect the folding/stability of the protein.  In my opinion, this variant is better classified as likely pathogenic (with criteria PM1, PM2, PP2, PP3) as opposed to VUS.
    • Our answer: the issue has been addressed; a paragraph has been added about its classification as likely pathogenic.
  • Patient 3.  The Y41del mutation in EGLN1 deletes Tyr-41.  This amino acid resides in the zinc finger of PHD2, and a Y41C mutation associated with erythrocytosis was previously reported to impair zinc finger function (PMC6161772).  This variant should therefore be classified as likely pathogenic (with criteria PM1, PM2, PM4, PM5) as opposed to VUS.
    • Our answer: the issue has been addressed; the variant has been classified as likely pathogenic.
  • All patients.  Are EPO values known for any of the patients?  If so, they should be reported.  This would be particularly helpful in understanding aetiology in those patients with two heterozygous mutations.  For example, patient 5 has mutations in the JAK2 and EPAS1 genes.  Erythrocytosis associated mutations in JAK2 are often associated with low EPO, whereas those in EPAS1 are often associated with increased EPO.  Hence, an EPO level in this patient may help in distinguishing the relative importance of the two mutations observed.
    • Our answer: the issue has been addressed; EPO values are known in all patients and have been added to the table. Additionally, they have also been described in the text.

Reviewer 2 Report

The authors report NGS sequencing results from a cohort of 34 patients identified as having unexplained erythrocytosis. Six of them (18%) had variants of unknown significance.
Major point:
- the list of variants consists of polymorphisms and/or variations with either no or unknown functional impact, making the results of little interest. In particular, the JAK2 G571S variant is known to be neutral. The VHL variant is a polymorphism....The EGLN1 variant Y41del is located in the zinc finger domain, thus it could be interesting to knwo wheter this variant is also noted in relatives.
- a fairly simple way to check if the variants have a functional impact is to study the segregation of the gene in relatives: has this study been repeated in patients with the variants?
- The authors mention erythrocytosis, but do not give the isotopic measurement values of the red cell mass, which alone can affirm the existence of true polycythaemia: can they detail these parameters for the patients?

In conclusion, the data presented are too preliminary to be relevant, further work is still needed.

Author Response

  • The list of variants consists of polymorphisms and/or variations with either no or unknown functional impact, making the results of little interest. In particular, the JAK2 G571S variant is known to be neutral. The VHL variant is a polymorphism. The EGLN1 variant Y41del is located in the zinc finger domain, thus it could be interesting to know whether this variant is also noted in relatives.
    • Our answer: In our opinion, The JAK2 1711G>A, p.(Gly571Ser) variant cannot be definitively described as neutral. This variant has already been described in the literature in a somatic context in MPN, not as clearly pathogenic. In vitro assays did not reveal any changes in activity for this variant compared to the wild type (ACMG criterion BS3). The clinical significance of this variant is therefore unclear. It was seen in triple negative MPN, and it lies in the JH2 inhibitory domain, possibly leading to constitutive JAK2signaling, but with reduced penetrance as shown in the literature.
  • A fairly simple way to check if the variants have a functional impact is to study the segregation of the gene in relatives: has this study been repeated in patients with the variants?
    • Our answer: According to Swiss law, genetic testing of patients and their relatives is strictly regulated. The clinical reality is that the feasibility of testing the relatives is doubtful in a specialized clinic as they need to be formally referred for the investigation, meaning they need to have a phenotype (polycythemia/erythrocytosis) confirmed by their general practitioner. Additionally, the relatives need to be willing to undergo the testing. As they are not known to our hematology department, we are not allowed to contact them in line to personal data protection and doctor-patient confidentiality law. We are therefore not able to perform this testing in the relatives.
  • The authors mention erythrocytosis, but do not give the isotopic measurement values of the red cell mass, which alone can affirm the existence of true polycythemia: can they detail these parameters for the patients?
    • Our answer: the issue has been addressed; this data has been added in the text and in the table. All patients in whom erythrocyte to plasma volume was measured (4/6, 66%) had absolute erythrocytosis, meaning increased erythrocyte volume. Additionally, values for RBC have been added in the table where available.
  • In conclusion, the data presented are too preliminary to be relevant, further work is still needed.
    • Our answer: We cannot agree that the data is too preliminary to be relevant. This study reports molecular findings in patients with the phenotype of polycythemia/erythrocytosis. To our knowledge, there are not many studies linking the molecular findings with the clinical picture. The advantage of this study is the highly selected patient cohort. We do agree that further work is still needed, however as mentioned by another reviewer, this manuscript adds to the growing literature describing mutations in erythrocytosis-associated genes.  

Reviewer 3 Report

In their paper the authors present the results of screening a cohort of patients with unexplained erythrocytosis using a 13 gene NGS panel that they designed in house. They uncovered 8 novel variants in 6 patients out of a total of 34. However the vast majority of the cohort did not possess any variants that would be considered to be erythrocytosis causing.

  1. Abstract, Line 15 – since the patients had a heterozygous mutation of unknown significance (especially patient 4) the authors state that the patients were diagnosed with erythrocytosis. Is the erythrocytosis confirmed by the presence of the variant of unknown significance or by the laboratory indices?
  2. The genes chosen to be screened have been reported previously associated with erythrocytosis.
    1. Did the authors consider screening promoter regions in their candidate genes, other than EPO?
  3. Methods, line 94 – “All patients were JAK2 negative”
    1. Do the authors mean V617F and exon 12 negative
  4. Was Sanger sequencing used to confirm the variants detected by the NGS panel
  5. EPO 3` UTR variant (c.*656G>A)
    1. Can the authors check if this variant is cited in gnomAD
      1. If so does this affect the variant classification?
    2. Line 121 – correct “coded sequence” to “coding sequence”
  6. Are the EGLN1 p.Leu363Arg and p.Tyr41del variants in regions with important functions or highly conserved?
    1. Are there other reported erythrocytosis causing mutations in these regions?
  7. The EPAS1 p.Gly156Trp is not present in the oxygen-dependent degradation domain of EPAS1 but is there any information to indicate that is a highly conserved region and has been assigned a specific function.
  8. Did the authors confirm whether the different JAK2 and SH2B3 variants detected were somatic or constitutional?
    1. JAK2 p.Pro390Leu is located in exon 9 but the authors state they sequenced exons 12-16 in the Methods. Can the authors clarify this discrepancy?
  9. For the six patients reported with variants were EPO levels measured or available?
    1. If so were these levels compatible with the identified defect in the stated gene?
      1. For example, for the EPAS1 and EGLN1 variants, was the EPO level in the normal range or elevated?
  10. In the era of genomics and metabolomics, can the authors in their Discussion be more speculative about other potential genes or mechanisms that could cause unexplained erythrocytosis

Author Response

  • Abstract, Line 15 – since the patients had a heterozygous mutation of unknown significance (especially patient 4) the authors state that the patients were diagnosed with erythrocytosis. Is the erythrocytosis confirmed by the presence of the variant of unknown significance or by the laboratory indices?
    • Our answer: According to WHO 2016 PV diagnostic criteria, Erythrocytosis is defined as: Hemoglobin >16.5 g/dL in menand >16 g/dL in women, or hematocrit >49% in men and >48% in women, or red cell mass >25% above mean normal predicted value. This explanation has been added in the Methods (line 67).
  • The genes chosen to be screened have been reported previously associated with erythrocytosis. Did the authors consider screening promoter regions in their candidate genes, other than EPO?
    • Suggestion has been added in the conclusions
  • Methods, line 94 – “All patients were JAK2 negative”. Do the authors mean V617F and exon 12 negative?
    • Our answer: the issue has been addressed as described above.
  • Was Sanger sequencing used to confirm the variants detected by the NGS panel?
    • Our answer: All mentioned variants were confirmed by Sanger sequencing. This explanation has been added in the text.
  • EPO 3` UTR variant (c.*656G>A). Can the authors check if this variant is cited in gnomAD. If so, does this affect the variant classification?
    • Our answer: The variant is not found in gnomAD. The region is out of the EPO gene region, but covered in gnomAD.
  • Line 121 – correct “coded sequence” to “coding sequence”
    • Our answer: the issue has been addressed and corrected.
  • Are the EGLN1 p.Leu363Arg and p.Tyr41del variants in regions with important functions or highly conserved? Are there other reported erythrocytosis causing mutations in these regions?
    • Our answer: The variant detected here is not listed in the following databases: HGMDprofessional, ClinVar, dbSNP, LOVD, gnomAD (ACMG criterion PM2). It can, however, be located in the catalytic domain of the EGLN1 protein spanning amino acids 291-392 (ACMG criterion PM1). Computational prediction tools indicate a deleterious effect (ACMG criterion PP3). The variant has been evaluated as likely pathogenic according to the ACMG criteria of PM1, PM2, PP2 and PP3. There are known variants causing erythrocytosis in this region, although they are rare.
  • The EPAS1 p.Gly156Trp is not present in the oxygen-dependent degradation domain of EPAS1 but is there any information to indicate that is a highly conserved region and has been assigned a specific function.
    • Our answer: The clinical significance of this variant is unclear. It was seen in triple negative MPN, and it lies in the JH2 inhibitory domain, possibly leading to constitutive JAK2 signaling, but with reduced penetrance as shown in the literature.
  • Did the authors confirm whether the different JAK2 and SH2B3 variants detected were somatic or constitutional?
  • Our answer: This is in progress, but has not been finalized yet.
  • JAK2 p.Pro390Leu is located in exon 9 but the authors state they sequenced exons 12-16 in the Methods. Can the authors clarify this discrepancy?
    • Our answer: this issue has been addressed in the text; exons 9-16 were sequenced.
  • For the six patients reported with variants were EPO levels measured or available? If so were these levels compatible with the identified defect in the stated gene?
    • Our answer: the issue has been addressed; EPO levels are known in all patients and were normal. As one of the reviewers mentioned, erythrocytosis associated mutations in JAK2 are often associated with low EPO, whereas those in EPAS1 are often associated with increased EPO. However, in this patient with normal EPO level, this laboratory test is not very helpful in distinguishing the relative importance of the two mutations observed. Unfortunately, the EPO measurement is not helpful nor compatible with the identified defects in the stated genes in our cohort of patients.

Round 2

Reviewer 2 Report

The answers of the authors are not sufficient to improve the quality of the article which does not bring anything in terms of scientific knowledge to the molecular anomalies of idiopathic erythrocytosis. Only the addition of the red cell mass in 4 of the 6 patients brings an essential data for the qualification of true polycythemia. It is however regrettable that the data of red cell mass of the whole cohort (n= 34) is not known: indeed, to carry out a long and expensive examination (NGS) in 34 patients for whom the diagnosis of true polycythemia is not verified by the measurement of the red cell mass is rather surprising. Experience shows that the hemoglobin and hematocrit criteria used by the WHO are only valid for polycythemia vera, and not for all erythrocytosis: indeed, the existence of high hemoglobin and hematocrit values, associated with the presence of the JAK2 mutation, makes the diagnosis of polycythemia vera very likely. However, this is not the case for other erythrocytoses, for which the first diagnostic step is to check for the existence of true polycythemia.
Furthermore, the JAK2 G571S variant is neutral (cf Irina Panovska-Stavridis Clinical Lymphoma, Myeloma & Leukemia,2016 Vol. 16, No. 5; Burak Bahar, Leukemia Research Reports 6 (2016) 27-28).
Regarding the legislation for genetic analysis of relatives, it is identical in most countries, and does not prevent contacting the propositus informing him of the usefulness of having a full blood count performed for relatives, and organizing a family segregation study: this work is done daily in constitutional genetic laboratories. This is a tedious task, but necessary for the characterization of the detected variants.